# The IL-33/ST2 Pathway in Cerebral Malaria

**DOI:** 10.3390/ijms232113457

**Published:** 2022-11-03

**Authors:** Corine Glineur, Inès Leleu, Sylviane Pied

**Affiliations:** Center for Infection and Immunity of Lille-CIIL, Institut Pasteur de Lille, CNRS UMR 9017-Inserm U1019, University Lille, 59019 Lille, France

**Keywords:** IL-33, ST2, endothelial, astrocytes, central nervous system, blood–brain barrier, *Plasmodium*, red blood cells, inflammation, cerebral malaria

## Abstract

Interleukin-33 (IL-33) is an immunomodulatory cytokine which plays critical roles in tissue function and immune-mediated diseases. IL-33 is abundant within the brain and spinal cord tissues where it acts as a key cytokine to coordinate the exchange between the immune and central nervous system (CNS). In this review, we report the recent advances to our knowledge regarding the role of IL-33 and of its receptor ST2 in cerebral malaria, and in particular, we highlight the pivotal role that IL-33/ST2 signaling pathway could play in brain and cerebrospinal barriers permeability. IL-33 serum levels are significantly higher in children with severe *Plasmodium falciparum* malaria than children without complications or noninfected children. IL-33 levels are correlated with parasite load and strongly decrease with parasite clearance. We postulate that sequestration of infected erythrocytes or merozoites liberation from schizonts could amplify IL-33 production in endothelial cells, contributing either to malaria pathogenesis or recovery.

## 1. Introduction 

### 1.1. Cerebral Malaria (CM)

Malaria affects an estimated 241 million individuals worldwide and contributes to about 627,000 deaths per year mainly from cerebral malaria (CM) in children below five years in Sub-Saharan Africa (Report OMS 2021). CM is a complex neuroinflammatory syndrome caused by *Plasmodium* infection. Patients generally suffer from a diffuse encephalopathy associated with a decrease of consciousness and a deep coma can precede death [1,2]. Currently there is no effective treatment. 

### 1.2. Immunopathological Mechanisms Associated to CM

Although the pathogenesis of CM is not completely elucidated, brain inflammation appears to be triggered by local events mediated by the sequestration of parasitized erythrocytes in the microcirculation and infiltrating immune cells. These events are preceded and amplified by the systemic secretion of pro-inflammatory cytokines by astrocytes and microglia cells upon interaction with the parasite. Those cytokines increase the permeability of the blood–brain barrier (BBB) and promote T cells and inflammatory monocytes infiltration which results in brain œdema, axonal damage, and myelin loss [3,4]. Experimental mouse models which utilize *Plamodium berghei* ANKA (*Pb*A) infection of C57BL/6 mice have shown that pathogenic CD4^+^ and CD8^+^ T cells, detected in the brain at the onset of neurological symptoms, play a role both at local and systemic levels by contributing to the high levels of TNF-α, IFN-γ and CXCL10 in circulation [5,6,7]. The development of CM is associated with recruitment in the brain of CD8^+^ T cells identified as one of the critical effector cell types mediating human cerebral malaria (HCM) and experimental cerebral malaria (ECM) [5,8]. These cells produce perforin which is implicated in the lytic granule exocytosis pathway used to kill targets. Perforin is also required to initiate the œdema occurring during ECM which also constitute a leading indicator associated with fatal HCM [9]. This is corroborated by the fact that perforin-deficient mice are protected from ECM despite brain infiltration of CD8^+^ T cells equivalent to that in susceptible mice. Moreover, perforin promotes the cerebral endothelial cell tight-junction disruption during *Pb*A infection [9]. Therefore, disruption of the BBB remains a key feature of the entire process that contributes to the neuropathology. 

### 1.3. Blood–Brain Barrier in Cerebral Malaria

The BBB is comprised of endothelial cells forming a continuous barrier through tight junctions, pericytes, a basement membrane and astrocytes, which are in direct contact with neurons and microglia [10]. This composition is critical to minimize local inflammation and neuronal damage, and dysfunction of the BBB appears to be associated with CM progression [3]. Blood–cerebrospinal fluid (B-CSF) barrier which also protects the CNS against harmful substances, is another site disrupted during ECM caused by *Plasmodium berghei* ANKA [11]. Cytoadherence of infected red blood cells (iRBCs) to endothelial cells (ECs) has been reported to cause over-expression of inflammatory cytokines [12] alter vascular integrity [13] and trigger ECs’ apoptosis [14,15,16]. Immunohistochemistry of brain sections derived from fatal pediatric CM cases indicated BBB impairment in areas containing sequestered *P. falciparum*-infected erythrocytes, with focal loss of endothelial intercellular junctions which results in œdema [17]. Microscopic data revealed a redistribution of the tight junction proteins occludin, vinculin, and zonula occludens 1, which are central to BBB integrity [17]. Even controversial, endothelial cell apoptosis has been suggested as a mechanism of increased endothelial permeability [18,19]. Although proximity of iRBCs to endothelial cells is important to activate a proinflammatory response and induce endothelial permeability, cytoadherence per se may not be crucial [20,21,22]. However, iRBCs cytoadherence to ECs is essential to the survival of the parasites because it allows them to evade clearance by the spleen. The sequestration of iRBCs leads to obstruction of microcirculatory blood flow, with resulting tissue hypoxia, metabolic disturbances, and organ dysfunction [23]. In addition, EC dysfunction may result from the release of hemozoin, a by-product of intraerythrocytic parasite-mediated hemoglobin catabolism [24]. 

## 2. IL-33

Interleukin-33 (IL-33) is a member of the interleukin-1 (IL-1) family that signals via the ST2 receptor (also called IL1RL1) [25]. The *St2* gene encodes two protein isoforms: ST2, a transmembrane receptor; and a secreted soluble ST2 form (sST2), which acts as decoy receptor for IL-33 (Figure 1).  Via binding to ST2, IL-33 signals recruit myeloid differentiation primary response protein 88 (MyD88) and IL-1 receptor associated kinases, leading to the activation of NF-kB, mitogen-activated protein kinase (MAPK) and PI3K/Akt pathways [26,27,28,29].

IL-33 is expressed constitutively in barrier tissues, lymphoid organs, and embryos, and can be induced in inflamed tissue [30,31]. IL-33 is mainly present in the nuclei of structural cells. In humans, the endothelium constitutes the major cellular source of IL-33 during steady state [32,33]. In contrast, in mice, it has been reported that Il-33 mRNA is not constitutively expressed in endothelial cells [31,34]. However, constitutive IL-33 expression has been observed in cultured primary pulmonary endothelial cells purified from murine lungs probably due to differences in proliferative properties of purified cells compared to steady state cells in mouse [29]. 

### 2.1. Nuclear and Extracellular IL-33

The IL-33 protein comprises two domains: the C-terminal domain containing IL-1 family member homology and mediating the extracellular, ST2-dependent effects of IL-33 [25], and the N-terminal domain targeting IL-33 to the nucleus where it binds to chromatin at the surface of the nucleosome by docking to the pockets of histone H2A–H2B dimer (Figure 1) [35]. Although this binding has been shown to confer a potent transcriptional repressor capacity in an artificial gene reporter assay [36], it has been reported that the nuclear IL-33 has no influence on the global transcriptome or proteome of cultured human epithelial or endothelial cells, respectively [37,38]. In fact, the nuclear restriction and heterochromatin enrichment of IL-33 are post-translational mechanisms that regulate the release from necrotic cells and ST2-mediated bioactivity of IL-33 (Figure 1) [37]. It is now clear that chromatin components or nucleosomes are important damage-associated molecular pattern (DAMPs) that induce proinflammatory signaling when released into the extracellular environment [39]. Interestingly, Luzina I.G. et al., demonstrated that full-length IL-33 (FL-IL-33), in contrast to the mature form (mIL-33), is functionally active in vivo in an ST2-independent fashion [40]. The FL-IL-33 intratracheally instilled in mice induced inflammatory cells’ infiltration in their lungs independently of ST2, whereas mIL-33 required ST2 to induce Th2-associated effects. Detailed mechanistic explanations for the ST2-independent effects of FL-IL-33 will require further analysis but we propose that FL-IL-33 complexed with chromatin could act as DAMPs and activate Toll-like-receptors (TLRs) (Figure 1). 

### 2.2. Post-Translational Regulation of IL-33

Processing of IL-33 by proteases is crucial to regulate its bioactivity. IL-33 cleavage by caspase-1, caspase-3 and caspase-7 during apoptosis, generates two inactive IL-33 products (Figure 1) [41,42,43]. Consistently, the administration of IL-33 in caspase-1-deficient mice greatly enhanced allergen-induced eosinophilic inflammation as compared with that in wild-type (WT) mice [44]. Moreover, Cayrol et al. have recently shown that FL-IL-33 is cleaved by proteases derived from various environmental allergens (fungi, house dust mites, cockroaches, and pollens) and that the cleaved short mature forms of IL-33 are potent inducers of allergic airway inflammation (Figure 1) [45,46]. In addition, six distinct inflammatory serine proteases were shown to cleave human IL-33 precursor, including neutrophil cathepsin G, elastase and proteinase 3 (PR3), as well as mast cell chymase, tryptase and granzyme B (Figure 1) [46,47]. These findings suggested that the bioactivity of IL-33 is finely regulated by endogenous and exogenous proteases. In fact, we recently published that, in murine primary endothelial cells, the IL-33 protein level is significantly increased when the allergen-induced Fas signaling pathway is inhibited by a caspase-8 inhibitor [29]. These results suggest that the allergen-induced Fas pathway regulates intracellular IL-33 protein level by increasing caspases’ activity. IL-33 bioactivity is also controlled by the binding of IL-33 to its receptor which is abolished by oxidation of four critical cysteine residues located in the IL-1-like cytokine domain of IL-33 (Figure 1). The formation of two disulfide bridges in IL-33 results in an extensive conformational change that prevents the ST2 binding site [48].

### 2.3. IL-33 in Immune Responses

IL-33 functions as an alarmin cytokine released upon tissue injury/cell death to alert immune cells [49,50]. IL-33 amplifies both Th1- and Th2-type responses through its activity on different ST2-expressing immune cells driving the production of Th1 or Th2-associated cytokines. (Figure 2) [51,52]. Even though IL-33 was initially recognized as a driver of Type 2 immunity, it is now clear that cytotoxic T cells, Th1, and Treg subsets can respond to this cytokine in certain contexts [53,54,55,56]. Notably, IL-33 can directly activate CD4^+^ and CD8^+^ T cells and enhance cytotoxic T lymphocytes responses [53,57,58].

However, IL-33 activity is dictated by its cellular source. Using mouse models characterized by conditional deletion of IL-33 in dendritic cells (DCs), or epithelial cells, Hung L-Y et al. demonstrated that epithelial IL-33 stimulates group 2 innate lymphoid cell (ILC2)–driven type 2 immunity and parasitic helminths’ clearance [59]. In contrast, IL-33 derived from DCs suppresses host-protective inflammatory responses. Interestingly, DCs can export IL-33 from the cytoplasm to the extracellular space through the transmembrane pore-forming protein perforin-2 as the absence of perforin-2 in DCs reduces the IL-33 level in supernatant and impairs their role on Treg growth.

IL-33 also impacts a wide range of immune cells and influences many facets of both innate and adaptive arms of the immune system [60] (recapitulated in Figure 2). Tissue–resident immune cells B cells are among the IL-33 main targets. IL-33 enhances B cells proliferation and IL-5, IL-13 and IgM secretion [25,61]. IL-33 is also a potent inducer of mucosal IgM^+^ IL-10-producing regulatory B cells [62] in mice, and IL-33 level correlates with increased plasma total IgM and auto-reactive antibodies in rheumatoid arthritis patients [63].

### 2.4. IL-33 in the Central Nervous System

Astrocytes, oligodendrocytes, neurons and microglia constitute regulatory cells within the central nervous system (CNS). Astrocytes are crucial regulators of innate and adaptive immune responses in the injured central nervous system and oligodendrocytes act mainly as supportive cells to allow correct synaptic and axonal function. Neurons are primarily involved in transfer information throughout the CNS. Microglia are primarily involved in cell processes that cause the release of pro-inflammatory cytokines, chemokines, and oxidative stress molecules. Throughout the lifespan, behavioral experiences can change neuronal structure and synaptic efficacy, remodel vasculature and glial processes, and alter the rate of neurogenesis [64]. 

In humans and mice, both the IL-33 protein and its mRNA are constitutively expressed at high levels within the brain and spinal cord tissues [52]. Around 33% of isolated brain cells of naïve mice are IL-33-positive [65,66] and IL-33 mRNA expression level is higher in the brain than in any other tissues and organs tested [25]. According to the human protein atlas, IL-33 mRNA is also expressed in the choroid plexus cells which form the B-CSF barrier, suggesting a role of this cytokine in this barrier. The abundance of IL-33 in the CNS suggests that IL-33 plays a role in the regulation and/or maturation of neuronal circuits. However, how IL-33 functions in the CNS is still under investigation. IL-33 is expressed in various brain regions including the corpus callosum and secondary motor cortex, the medial prefrontal cortex, the periventricular hypothalamic nucleus, and the amygdala. Via its ST2 receptor, IL-33 is thought to have a role in neuro-inflammatory diseases [67]. Several studies reported the involvement of IL-33/ST2 signaling in the pathogenesis of Alzheimer’s disease (AD) [68], multiple sclerosis (MS) [69], experimental autoimmune encephalomyelitis [70], experimental cerebral malaria (ECM) [71], chronic pain [72], intracranial hemorrhage (ICH) [73], and stroke [74]. Interestingly, increased concentrations of IL-33 has been reported in both serum and CSF samples from patients with MS [69]. Neuroprotective effects of IL-33 have been demonstrated in AD, MS, chronic pain, ICH and stroke. In addition, in different mouse models, exogenous IL-33 administration decreases CNS damage and ameliorates neurological deficits by promoting systemic anti-inflammatory responses and increasing M2-type macrophages and microglia in the CNS [75] by inducing a shift from Th1 to Th2 and suppressing Th17 responses [76], or by upregulating the expression of IL-10 and other M2 genes in microglia [77]. 

In the CNS, ST2 is mainly produced by neurons, astrocytes and microglia [75,78]. Moreover, IL-33 is constitutively expressed in oligodendrocytes and astrocytes [65,77,79]. IL-33 engagement by microglia enhances the production of chemokines such as RANTES, CCL2, MIP-1α, and CXCL10, of cytokines such as IL-1β, TNF-α, IL-10, and Th2 cytokines such as IL-13 as well as that of nitric oxide (Figure 3) [77]. Astrocyte-derived IL-33 communicates with microglia to tune synapse engulfment during neural circuit maturation and remodeling [80]. Yet, IL-33 is particularly expressed in a molecularly distinct subset of neurons enriched for markers of synaptic plasticity [81]. Finally, IL-33 deficiency has also been associated with anxiety-related behaviors [30]. 

## 3. IL-33 in Cerebral Malaria

Plasma IL-33 concentrations are higher in patients with severe malaria than in patients with uncomplicated malaria and infection-free controls [82], suggesting that IL-33 is associated with the onset of severe malaria. Of note, the levels of sST2 which acts as a decoy receptor for IL-33 are elevated in plasma and CSF of children with CM. The sST2 levels correlate with markers of endothelial activation and neuronal damage and predict neurocognitive impairment [83]. Moreover, in a model of *Pb*A-induced ECM, IL-33 protein level in brain doubled upon *Pb*A blood-stage or sporozoite infection without any significant increase of its mRNA level [84]. However, the exact role of IL-33 in these settings is poorly understood. 

Given its high expression level in the brain and particularly in cells such as astrocytes and oligodendrocytes and supposedly endothelial cells, IL-33 could play a crucial role in CM.

### 3.1. IL-33 Immune Regulation in Cerebral Malaria

In humans and mice, CM is marked by the sequestration of *Plasmodium-*infected red blood cells (iRBCs) in the brain microcirculation, resulting in blood vessel occlusion, inflammation and brain swelling [85]. This sequestration occurs because iRBCs can bind to two proteins on the surface of the endothelium of brain blood vessel: intercellular adhesion molecule-1 (ICAM-1), and endothelial protein C receptor (EPCR) [86]. RBCs are one of the major sources of plasma IL-33, and IL-33 levels are considerably increased in supernatants from lysed RBCs [87]. Moreover, IL-33 induces adhesion molecules’ expression and activates inflammation in human and murine endothelial cells, and thus promotes the adhesion of human leukocytes to monolayers of human endothelial cells [29,88]. In ECM, T lymphocyte recruitment and activation in the brain are essential for pathogenesis, in particular CD8^+^ effector T cells [89]. It has been proposed that endogenous IL-33 may promote cytotoxic CD8^+^T-cell responses during sporozoite *Pb*A-driven ECM development.

### 3.2. IL-33, Autophagy and Cerebral Malaria

Multiple molecular mechanisms involved in CM, include not only the disruption of BBB and activation of inflammatory response but also the degradation of parasite microvesicules (*Pb*A-MVs) transferred in astrocytes [90]. Autophagy is a self-degradation process activated in response to stress, and present in all mammalian cells and tissues, including the CNS [91]. This process directs damaged intracellular material to the lysosomes and involves a network of at least 16 proteins that promote the autophagosome formation [92]. Conversion of cytoplasmic microtubule-associated protein 1A/1B-light chain 3 (LC3) -I form to the membrane-bound LC3-II is a useful biomarker to detect autophagy. Leleu I. et al. recently published that LC3-Associated phagocytosis (LAP) activation in astrocytes could participate in the processing of *Pb*A antigens and their presentation by major histocompatibility complex class I (MHC-I) molecules [93]. This leads to the induction of proinflammatory response of astrocytes and the recruitment of a selective repertoire of CD8^+^ T cells infiltrating the brain during CM [5,94]. 

Interestingly, IL-33 modulates autophagy in a cell-context or pathology-dependent manner. For instance, IL-33 promotes the macrophages autophagy in the setting of experimental colitis and IL-33 macrophagic activity is associated with the conversion of LC3-I to LC3-II [95]. In a mouse model of liver injury, IL-33 exerts protective effects on hepatocytes through the activation of autophagy and functions as an innate immunity regulator mediating macrophage polarization [96]. In contrast, in a neonatal seizure model, IL-33 treatment inhibits autophagy, apoptosis, neuroinflammation and improves neurologic outcome through ST2 [73].

### 3.3. Proposed Role of IL-33 in CM-Associated Endothelial Disruption

Studies have shown that the human perforin secreted by CD8^+^ T cells causes endothelial disruption and fatal œdema [9]. Indeed, *Plasmodium* perforin-like proteins can form pores on human RBCs leading to their lysis [97]. Since murine perforin-2 has been shown to be necessary to release IL-33 and promote its function [59], it would be interesting to investigate the role of IL-33 in CD8+ T cells perforin-induced endothelial disruption. 

During CM, parasite histones from merozoites released in the proximity of endothelial cells during schizogony of cytoadherent iRBCs have been proposed to exert a disruptive effect on endothelial barrier function through a charge-related mechanism [98]. Indeed, parasite histones are able to activate innate receptors and signaling pathway involving Src family kinases and p38 MAPK [98]. Host cell-derived histones may also be released as a result of cell injury. In agreement with these observations, nucleosome levels were markedly elevated, in patients plasma during severe malaria, compared with mild infection and correlated positively with disease severity [98,99]. Recent data pointed that chromatin regulates IL-33 release in the form of high-molecular weight complexes [37]. The fact that elevated serum histone levels have been shown to be implicated in proinflammatory responses and BBB disruption in malaria may question the role of IL-33 in these particular functions of host and parasitic histones. 

### 3.4. IL-33 in Experimental Models of Cerebral Malaria

The majority of IL-33 and malaria studies were conducted in *Pb*A-infected mice either in WT mice or in IL-33- or ST2-deficient mice (Table 1). Surprisingly, while administration of recombinant IL-33 in *Pb*A-infected C57BL/6 mice attenuated the development of ECM, it also reduced symptoms on ST2-deficient mice. Another study showed that IL-33-deficient mice have a similar survival and parasitemia than WT mice [100]. The reasons of these contradictory results are still debated. As a multifunctional cytokine, IL-33 appears to have both beneficial and pathological roles particularly in lung responses to pathogens [50]. While IL-33 can cause inflammation-induced airway tissue damage, it can also promote tissue repair after injury. The underlying mechanisms appear to involve immune cells as well as non-immune lung cell and different signaling pathways.

In CM, the beneficial effect of IL-33 administration has been attributed to induction of M2 polarization, reduction of inflammatory mediators and expansion of type-2 innate lymphoid cells (ILC2s) and regulatory T cells (Tregs) (Figure 3) [71]. Importantly, IL-33 has been shown to be a potent activator of ILC2s whose systemic expansion is shown to correlate with improved protection against CM. IL-33 could induce ILC2s to promote M2 activity in vitro and in vivo, which could promote Treg expansion. IL-33-mediated Treg function is important in preventing CM [71] and the Treg cell accumulation is promoted by a network of IL-33-responsive cell types, such as ILC2s and mast cells (MCs) [50,102]. The role of MCs in the outcome of CM severity may be different depending on IL-33. IL-33 and MCs can play an anti-inflammatory by promoting expansion of IL-10-producing Treg [102]. However, it has been reported that MCs-derived exosomes could worsen pathogenesis of ECM in mice [103]. Finally, it has been published that IL-33 administration improved the efficacy of the anti-malarial drugs and the effect of IL-33 is associated with decreased IL-1β production and NLRP3 inflammasome formation in the mice brain [104]. 

However, IL-33/ST2 pathway may also contribute to the exacerbated neuroinflammation during ECM development by inducing microglia to produce IL-1β that in turn may cause oligodendrocyte stimulation and IL-33 production (Figure 3) [101]. 

From ST2-deficient mice data set, we could conclude that IL-33 activity on ST2^+^ cells may contribute to the development of severe CM, while IL-33 activity may induce a therapeutic effect outside of the IL-33/ST2 axis. However, the amount of IL-33, its cellular location, the IL-33- and ST2-expressing cell types and stage of the disease IL-33 acts on, as well as the different receptors, ST2 or sST2, it binds to, may make the effects of IL-33 beneficial or deleterious during the pathology.

## 4. Conclusions

Here, we highlighted how IL-33 could influence cerebral malaria pathogenesis. To date, it is still unclear whether IL-33 has protective or destructive effects on malarial pathology. Future research should be devoted to deciphering the roles of IL-33 in neuroinflammatory pathways and in vascular permeability increase which leads to fatal œdema and in host-defense mechanisms such as autophagy or the production of auto-antibodies in the course of infection. The IL-33/ST2 pathway could be a pharmacological target of choice for the treatment of severe malaria. However, a better knowledge of early innate immune responses involving IL-33/ST2 signaling is required before developing new treatment for cerebral malaria targeting IL-33 modulation.

IL-33 is known to modulate IL-1β which is increased during CM [105] Although the role of IL-1β in CM is still poorly defined, several findings show that IL-1β secretion can be deleterious to the host. Postmortem analysis detected increased IL-1β in human brains with cerebral malaria [106]. In addition, IL-1β- deficient mice showed a better survival than wild type mice in ECM [107]. Thus, although the IL-33/ST2 pathway has been reported to be beneficial in ECM through expansion of ILC2, M2 macrophage and Treg [71], it can also be associated with a detrimental role via IL-1β. Therefore, new investigation for malaria therapeutics should consider IL-1β and IL-33 as potential targets. In this context, pharmacological inhibition of the IL-1 axis using existing biopharmaceuticals and administration of recombinant IL-33 could offer new prospects to treat cerebral malaria.

## Figures and Tables

**Figure 1 ijms-23-13457-f001:**
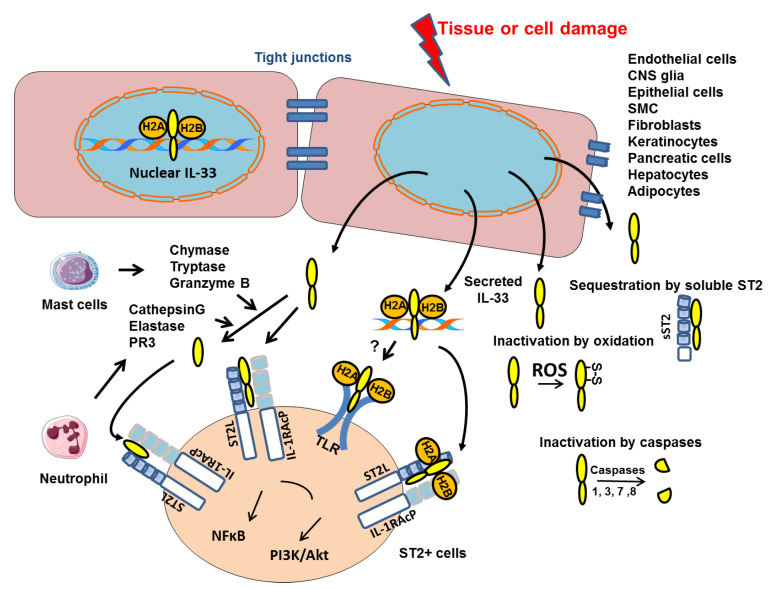
Mechanisms of intracellular IL-33 action and secretion. IL-33 is constitutively present in the nucleus of structural cells where it binds to chromatin by docking to the pockets of the histone H2A–H2B dimer. Upon cell or tissue damage, the IL-33 alarmin, is released extracellularly as a chromatin complex and is able to bind to receptors such as ST2 complexed to the coreceptor IL-1 receptor accessory protein (IL-1AcP) or supposedly TLR. IL-33 processing by inflammatory proteases produced by mast cells or neutrophils can generate mature forms with increased activity (up to 30-fold). Conversely, cleavage of the IL-1-like domain of IL-33 by caspases 1, 3, 7 or 8 could be an important mechanism of IL-33 inactivation during apoptosis. Extracellular IL-33 may also be sequestrated by soluble ST2 (sST2) or oxidized (formation of S-S disulfide bridges). All these inactivation mechanisms limit the range and duration of the ST2-dependent responses in vivo.

**Figure 2 ijms-23-13457-f002:**
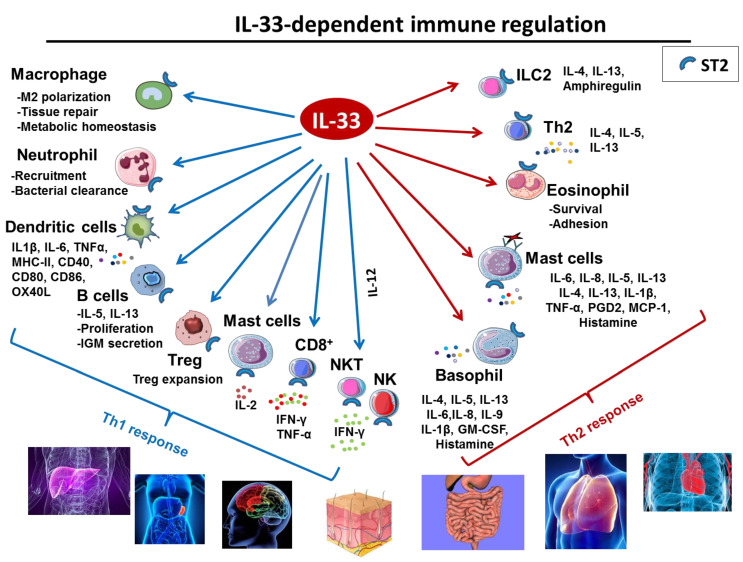
Role of IL-33 in immune response. IL-33 is a potent multi-organ pro-inflammatory cytokine acting on cells involved in Th1 orTh2 responses.

**Figure 3 ijms-23-13457-f003:**
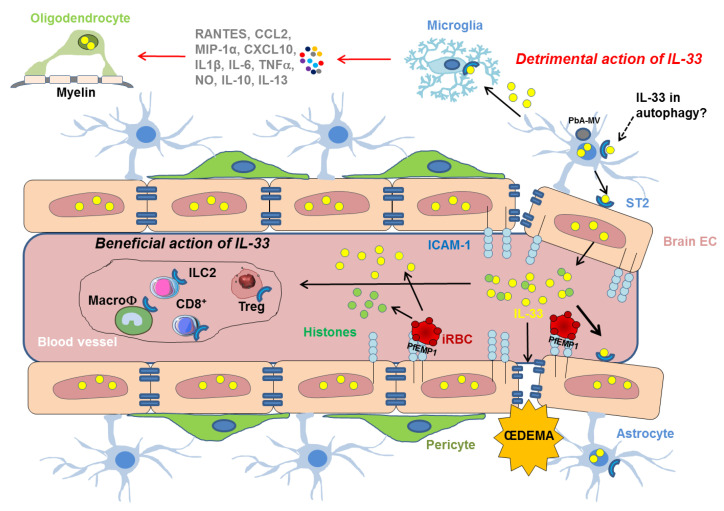
Mechanistic insights on the cellular and molecular players involved in the effect of IL-33/ST2 pathway, following the sequestration of iRBC in the vasculature of the BBB. IRBCs bind to ICAM-1 at the surface of endothelial cells via the *Plasmodium* erythrocyte membrane protein 1 (*Pf*EMP1). IRBCs may also release IL-33 (yellow circles) that can activate ICAM-1 expression on EC promoting the binding of iRBC. Histones (green circles) released by iRBC may synergize with IL-33. IL-33 drives the expansion of ILC2 cells, leading to the polarization of the anti-inflammatory M2 macrophages, which in turn expand Tregs giving a beneficial role of the IL-33/ST2 pathway in ECM. IL-33 may also have detrimental activities by inducing vascular permeablity responsible for œdema and targeting cells involved in the integrity of BBB, such as microglia which can then produce IL-1β resulting in neuroinflammation and IL-33 overexpression in oligodendrocytes. Autophagy pathways participate in the neuropathophysiological mechanisms of CM in driving the transfer of *Pb*A-microvesicles (*Pb*A-MVs-grey circle) and the induction the proinflammatory response in astrocytes. This pathway could also involve the IL-33/ST2 activity.

**Table 1 ijms-23-13457-t001:** Role of IL-33/ST2 pathway in PbA-infected mice models.

Mice	Role of IL-33	References
WT	Microglia activation/proliferation induced by IL-33 in CNS inflammation	[65,77]
	Effect of IL-33 mediated by systemic ILC-2, Th2, M2 and Treg	[71]
	In the brain during ECM, IL-33 mRNA expression not altered but IL-33 protein doubled	[84]
	Nuclear IL-33 detected by immunostaining in the brain during ECM.	[84]
	No circulating IL-33 detected in serum of *Pb*A-infected mice	[84]
	IL-33 increased in the spleen and lung after sporozoite infection	[84]
	Contribution of IL-33 to cognitive defects associated in ECM	[101]
	Early and direct role of IL-33/ST2 pathway in the exacerbated neuroinflammation during ECM	[101]
	IL-1β induced by IL-33/ST2 pathway triggers IL-33 expression in oligodendrocytes	[101]
	Exacerbated IL-33 expression by astrocytes and oligodendrocytes after *Pb*A-infection	[101]
	Decrease of IL-33 mRNA and upregulation of IL-33 protein in hippocampus, SVZ and frontal cortex in ECM	[101]
ST2-deficient	Reduction of cerebral inflammation	[84]
	Significant reduction of ECM symptoms	[84]
	Resistant to *Pb*A-induced neuropathology and improved survival	[84]
	Reduced brain sequestration of CD4+ T cells and CD8+ T cells	[84]
	Local expression of ICAM-1, CXCR3, and LT-α strongly reduced	[84]
	Parasitemia and brain parasite load similar to WT mice	[84]
	IFN-γ, TNFα and CXCL10 mRNA expression not altered in the brain	[84]
	No cognitive defect post-*Pb*A infection	[101]
	CXCL9, CXCL10, IL-1β drastically reduced	[101]
	Microglia activation after *Pb*A-infection	[101]
IL-33-deficient	Similar survival and parasitemia than WT mice	[100]
	Reduced anxiety-like behavior	[30]

## Data Availability

Not applicable.

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
