# Peer review of "The IL-33/ST2 Pathway in Cerebral Malaria"

_ijms, 2022, doi:10.3390/ijms232113457_

Round 1

Reviewer 1 Report

The paper entitled ‘The IL-33/ST2 pathway in cerebral malaria’ concerns a very interesting and important topic which is the pathophysiology of malaria infection with particular emphasis on the role played by IL-33 in cerebral malaria. The manuscript generally is well organized and the text is easy to flow through. In their review, the authors cite numerous works on the synthesis and physiological role of IL-33 as well as sometimes conflicting reports on the role of this cytokine in the pathogenesis of malaria. In my opinion, the paper fits well in the scope of the Journal and could be very interesting for the readers of IJMS.

Although the manuscript is quite long, I would suggest adding a chapter on the role of another important brain barrier in the pathophysiology of malaria: the blood-cerebrospinal fluid barrier. Blood-cerebrospinal fluid barrier is considered to be another important gateway through which parasites and other pathogens can enter the brain parenchyma. There are reports that the blood-cerebrospinal fluid barrier disruption occurs during experimental cerebral malaria. According to the human protein atlas IL-33 mRNA is also expressed in the choroid plexus cells, therefore gathering information about the potential role of this cytokine in the regulation of the blood-cerebrospinal fluid barrier could be very valuable.

Concluding, the article submitted for evaluation is valuable and after minor revision made it could be considered for publication in IJMS.

Author Response

Dear reviewer,

Thank you for careful revision of our review.

We added in abstract, in subsections 1.2 (Blood-brain barrier in cerebral malaria) and 2.4 (IL-33 in the central nervous system), sentences about blood–cerebrospinal fluid (B-CSF) barrier  in cerebral malaria and the potential role of IL-33 at this barrier.

Reviewer 2 Report

Glineur et al.  describe the recent advances regarding  the role of IL-33 and of its receptor ST2 in cerebral malaria, and in particular,  the role that IL-33/ST2 signaling pathway could play in brain blood barrier permeability. The manuscript is interesting and clear, providing information on the mechanism of cerebral malaria, the roles of IL-33 and IL-33/ST2. The figures regarding the mechanism of action and secretion of IL-33, the immune role of IL-33 and the roles of IL-33 in blood vessels and in  microglia are explicative and illustrate the text. The main results  regarding experimental cerebral malaria are included in a table. However, the part regarding the  contradictory effects of IL-33 could be improved, considering the putative roles of ST2 and soluble ST2. Could levels of IL-33 (high, low) or the stage of the pathology (early, late) influence the positive or negative role of IL-33 on cerebral malaria? Do the levels of soluble ST2 change in cerebral malaria? Maybe the Authors could include and discuss the recent results described by Fernarder et al. Elevated Plasma Soluble ST2 Levels are Associated With Neuronal Injury and Neurocognitive Impairment in Children With Cerebral Malaria. doi: 10.20411/pai.v7i1.499.

 There are few minor points that should be addressed before publication.

248 For instance, IL-33 promotes the macrophages autophagy in the setting of experimental colitis by promoting the conversion of LC3-I to LC3-II.

This sentence is not correct. The authors demonstrate that IL-33 promotes autophagy, and evidence it by measuring the conversing of LC3-I to LC3-II. This does not mean that IL-33 directly promotes the conversion of LC3-I. Please clarify and rewrite.

267 Recent data pointed that chromatin regulates IL-33 release in the form of high-molecular weight complexes.

Please include here reference 36 or other suitable reference

312 Altogether, the data suggest that IL-33 activity on ST2+ cells may contribute to the  development of severe CM, while IL-33 activity may induce a therapeutic effect outside  of the IL-33/ST2 axis.

The statement “IL33 activity may induce a therapeutic effect outside the IL-33/ST2 axis appears to be in contradiction with the precedent paragraph and figure 2. 288-90 : IL-33 could induce ILC2s to promote M2  activity in vitro and in vivo, which could promote Treg expansion. IL-33-mediated Tregfunction is important in preventing CM [70] and the Treg cell accumulation is promoted 290 by a network of IL-33-responsive cell types, such as ILC2s and mast cells. Figure 2 clearly indicates that mast cells and ILC2 express ST2. Could the Authors clarify this point?

Some typos/grammar mistakes should be corrected. These are some examples:

36  Experimental mouse models which utilizes: utilize

103 histones HA2-H2B dimer: histone

308 This pathway could also involved the IL-33/ST2 activity: involve

Author Response

Dear reviewer,

Thank you for careful revision of our review.

We added in chapter 3 (IL-33 in cerebral malaria) a sentence about sST2 levels in plasma and CSF of children with CM.

We also changed the improper sentence “IL-33 promotes the macrophages autophagy in the setting of experimental colitis by promoting the conversion of LC3-I to LC3-II” byIL-33 promotes the macrophages autophagy in the setting of experimental colitis and IL-33 macrophagic activity is associated with the conversion of LC3-I to LC3-II”.

We added a reference (37) after the sentence “Recent data pointed that chromatin regulates IL-33 release in the form of high-molecular weight complexes”.

We agree that the last sentence of the subsection 3.4 (IL-33 in experimental models of cerebral malaria) is confuse and over-reductionist. The conclusions are drawn from the ST2-deficient mice data. I have somewhat nuanced by adding different situations that could influence the direction that IL-33/ST2 can take, either beneficial or deleterious in CM.

Finally, we included  a brief paragraph on the potential dual role of mast cells in CM which is an emerging field under investigation in the subsection 3.4 (IL-33 in experimental models of cerebral malaria). We also changed figure 2 by incorporing MCs in Th1 response.

Typos/grammar mistakes have been corrected.